# An Efficient Real-Time FPGA-Based ORB Feature Extraction for an UHD Video Stream for Embedded Visual SLAM

Mateusz Wasala * , Hubert Szolc and Tomasz Kryjak *

Embedded Vision Systems Group, Computer Vision Laboratory, Department of Automatic Control and Robotics, AGH University of Science and Technology, Al. Mickiewicza 30, 30-059 Krakow, Poland; szolc@agh.edu.pl
* Correspondence: mateusz.wasala@agh.edu.pl (M.W.); tomasz.kryjak@agh.edu.pl (T.K.)

**Abstract:** The detection and description of feature points are important components of many computer vision systems. For example, in the field of autonomous unmanned aerial vehicles (UAV), these methods form the basis of so-called Visual Odometry (VO) and Simultaneous Localisation and Mapping (SLAM) algorithms. In this paper, we present a hardware feature points detection system able to process a 4K video stream in real-time. We use the ORB algorithm—Oriented FAST (Features from Accelerated Segment Test) and Rotated BRIEF (Binary Robust Independent Elementary Features)—to detect and describe feature points in the images. We make numerous modifications to the original ORB algorithm (among others, we use the RS-BRIEF instead of classic R-BRIEF) to adapt it to the high video resolution, make it computationally efficient, reduce the resource utilisation and achieve lower power consumption. Our hardware implementation supports a 4 ppc (pixels per clock) format (with simple adaptation to 2 ppc, 8 ppc, and more) and real-time processing of a 4K video stream (UHD—Ultra High Definition, 3840 × 2160 pixels) @ 60 frames per second (150 MHz clock). We verify our system using simulations in the Vivado IDE and implement it in hardware on the ZCU 104 evaluation board with the AMD Xilinx Zynq UltraScale+ MPSoC device. The proposed design consumes only 5 watts.

**Keywords:** Oriented FAST and Rotated BRIEF; ORB; feature points; feature extraction; FAST detector; BRIEF descriptor; UHD; 4K resolution; real-time; FPGA; Zynq UltraScale+





## 1. Introduction

Unmanned Aerial Vehicles (UAVs) (for the purposes of this work, we restrict this term to multi-rotor vehicles such as quad- and hexa-copters) have become very popular in recent years. They are currently used for a variety of tasks, such as filming, transporting small goods, exploring unknown spaces, or various types of inspections. Nowadays, one of the main trends in UAV development is autonomy, understood as the ability to carry out missions without (or with only minimal) operator involvement. This requires the vehicle to be equipped with several systems responsible for environment perception, navigation, and control. In addition, in some missions all calculations need to be performed using an on-board computer, for example, exploration of hard-to-reach areas where a high-throughput connection to a base station may be unavailable. Typical on-board computers for UAVs are characterised by a relatively limited computing power (due to weight and available energy), which poses a significant challenge for the developers of perception and navigation systems.

One of the previously mentioned problems concerns the exploration of an unknown space by a UAV without a GNSS (Global Navigation Satellite Systems) signal. In this task, the SLAM (Simultaneous Localisation and Mapping) algorithm can be used. It is based on the data acquired by the sensors available on board (e.g., cameras, inertial measurement unit—IMU) and determines the current position of the UAV, as well as generates a map of the unknown environment. When the GNSS signal is unavailable or disrupted, the vehicle

must perform a specific task based on the sensors it is equipped with. Among them, the camera deserves special attention. In particular, vision-based algorithms can be used to detect the displacement and rotation of the UAV, which can be used for its positioning. However, for this purpose, it is necessary to determine the geometric relationships between two images captured by the camera. The most common approach to obtain them is to use detection and description of feature points. There are many well-established algorithms for this task. Worth mentioning are: SIFT (Scale-Invariant Feature Transform) [1], SURF (Speeded-Up Robust Features) [2], and ORB (Oriented FAST and Rotated BRIEF) [3]. The detected points are then matched by comparing their descriptors, thus obtaining the corresponding locations in both images (e.g., corners of objects visible in both frames). This allows the desired geometric relationships to be determined. The accuracy of the obtained results depends on the quality of the details (colours, edges, shapes) visible in both input images.

Nowadays, UAVs are equipped with high-end 4K cameras. Due to very high resolution, the obtained video stream is characterised by a huge amount of high-quality details. This allows for a more accurate determination of geometric relationships. At the same time, the processing of 4K images requires more computational resources, especially for a real-time implementation (above 30 frames per second). This poses a particular challenge in case of embedded platforms typically used for UAVs. Heterogeneous Systems on a Chip (SoC) , which integrate different types of computing units, e.g., CPUs and FPGAs (Field-Programmable Gate Arrays), are exemplary solutions that enable high-resolution video stream processing with a relatively low power consumption.

In this paper, we present a hardware implementation of the ORB algorithm developed for the real-time implementation of the Visual SLAM algorithm for the Unmanned Aerial Vehicles. We have prepared a fully pipelined system operating in a parallel manner using a heterogeneous SoC FPGA device. The main contribution of our work includes the implementation of the FAST detector and the BRIEF descriptor for a 4K resolution video stream (UHD—Ultra High Definition, 3840 $\times$ 2160 pixels) @ 60 frames per second, 150 MHz clock. Due to the characteristics of the 4K video stream and the hardware platform used, we apply a vector data format (4 ppc—4 pixels per clock cycle), which imposes a different hardware architecture of the ORB algorithm. We also propose the first FPGA implementation of the *Fast Score* method. To our best knowledge, this is the first real-time implementation of the ORB algorithm for such a high-resolution video stream.

The remainder of this paper is organised as follows. Section 2 presents basic information about the ORB (FAST+BRIEF) algorithm. Then, Section 3 discusses the most important related work on hardware implementation of the algorithms considered in FPGA. The proposed system is described in Section 4, while the results obtained are summarised in Section 5. The article ends with conclusions and a discussion on future research.

## 2. Oriented FAST and Rotated BRIEF

In this section, we present a brief overview of the ORB algorithm used for feature point detection and description. This solution was proposed in 2011 in the article [3] as a kind of compromise between the SIFT and SURF algorithms. It performs as well as SIFT on the task of feature detection while being almost two orders of magnitude faster, and it is better than SURF, as it has higher detection quality and computational efficiency. The ORB algorithm consists of a FAST (Features from Accelerated Segment Test) oriented corner detector and a BRIEF (Binary Robust Independent Elementary Features) rotated feature descriptor.

When processing an image on a CPU, the ORB algorithm usually consists of the following steps:

1. Performing the fast accelerated segment test to determine the corners (FAST).
2. Filtration using non-maximum suppression (NMS).
3. Elimination of feature points for which it is not possible to determine the full context $31 \times 31$ pixels (px).

4.    Filtering of feature points and leaving only the best *N* points.
5.    Computation of the Harris score and re-filtering of feature points.
6.    Calculation of the orientation of the feature point (intensity centroid).
7.    Determination of the context $31 \times 31$ px and blurring with a Gaussian filter.
8.    Determination of the binary feature descriptor (rBRIEF).

### 2.1. Oriented FAST Feature Detector

The FAST algorithm is a corner detector that was originally presented in the paper [4] and developed further in the paper [5]. Its most important advantage is the high computational efficiency (compared to other algorithms, i.e., SIFT or SURF) while maintaining reasonable accuracy. For this reason, it is successfully used in real-time image processing applications on embedded platforms. Feature point detection is performed in the context of $7 \times 7$ px by comparing the brightness of a given image point with 16 pixels located in a surrounding Bresenham circle of radius 3, as shown in Figure 1. A centre point with intensity $I_p$ is considered a corner if the brightness of consecutive *n* pixels in the circle is simultaneously lower than $I_p - t$ or simultaneously higher than $I_p + t$, where *t* defines an intensity threshold. The authors of [4,5] showed in their study that the best quality results can be obtained by setting $n = 9$. It may be considered as a general statement (regardless of image resolution), as this depends primarily on the fixed radius of the used Bresenham circle.

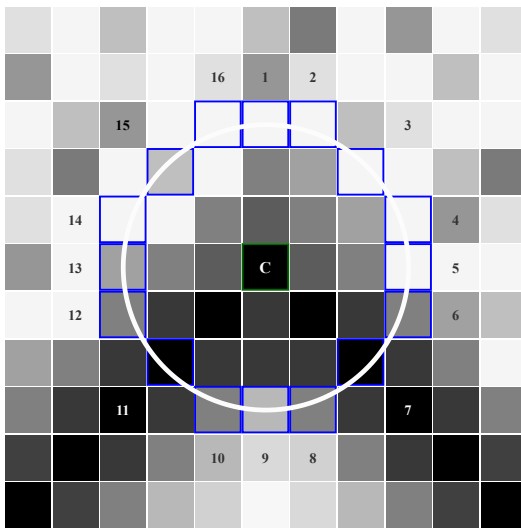

**Figure 1.** The Bresenham circle used in the FAST algorithm. Note that for $n = 9$ the centre point will be detected as a corner for a relatively wide range of possible thresholds *t*.

Furthermore, the algorithm assumes the calculation of a corner score to determine the stability of the detected feature point. For this purpose, depending on the particular implementation, one of several metrics is used, e.g., the Harris corner score, the sum of absolute differences between the candidate point and the pixels on the Bresenham circle [4] or a modification of the sum of absolute differences referred to as *Fast Score*.

The Harris corner score is determined by Equation (1). For each corner, the value of $H(x,y)$ is calculated based on the determinant and the trace of the autocorrelation matrix $M(x,y)$.

$$H(x,y) = \det(M(x,y)) - k * trace^2(M(x,y))$$

$$M(x,y) = h(x,y) \begin{bmatrix} I_x^2(x,y) & I_x(x,y)I_y(x,y) \\ I_x(x,y)I_y(x,y) & I_y^2(x,y) \end{bmatrix} \qquad (1)$$

where det is the matrix determinant, *trace* is the matrix trace, *k* is the constant value, $k \in [0.04; 0.06]$, $(x,y)$ are the image coordinates, $h(x,y)$ is the Gaussian function, $M(x,y)$ is the autocorrelation matrix, $I_x(x,y)$, $I_y(x,y)$ are the spatial derivatives.

The sum of the absolute differences is calculated using Equation (2). For each point $x \in \{1, \ldots, 16\}$ on the circle two sets: $S_{bright}$ and $S_{dark}$ are determined. The first set contains pixels greater than or equal to $I_p + t$, and the second pixels less than or equal to $I_p - t$. Then, the sum of absolute differences between the intensities of the pixels in each set and the intensity of the central pixel is determined, decreased by the threshold $t$. The final value of the measure is the maximum value of the two sums obtained.

$$V = \max \left( \sum_{x \in S_{bright}} \left| I_{p \to x} - I_p \right| - t, \sum_{x \in S_{dark}} \left| I_p - I_{p \to x} \right| - t \right) \tag{2}$$

$$S_{bright} = \left\{ x | I_{p \to x} \geq I_p + t \right\}$$
$$S_{dark} = \left\{ x | I_{p \to x} \leq I_p - t \right\} \tag{3}$$

The mentioned modification of the sum of absolute differences has been used in the OpenCV library [6]. In the documentation, this modification is referred to as *Fast Score*. For each set of 9 contiguous pixels from the Bresenham circle, the absolute differences in intensity between these 9 pixels and the central point are calculated. Then, the minimum of the 9 obtained values is determined. These operations are repeated for the remaining contiguous arc sets, obtaining 16 minimum values. The corner score of the feature point is the maximum of the 16 minimum values according to the Equations (4).

$$d^k = \left\{ |I_{p \to x} - I_p|, x \in \{1, \ldots, 9\} \right\}$$
$$d^k_{min} = min \left\{ d^k_i, i \in \{1, \ldots, 9\} \right\} \tag{4}$$
$$score = max \left\{ d^k_{min}, k \in \{1, \ldots, 16\} \right\}$$

where: $d^k$ are the difference values from one contiguous arc set of pixels, $I_p$ is the pixel value, $d^k_{min}$ is the minimum value from one contiguous arc set of pixels, *score* is the corner score, *Fast Score*.

The next step of the ORB algorithm is non-maximum suppression with a context $3 \times 3$ px. It allows to eliminate candidate points with a lower corner score and leave only the best N feature points.

However, the obtained corners are not invariant to the rotation angle of the image and its scale. To solve the first problem, the intensity centroid [7] as a measure of orientation was used. First, the moments of an image patch around a given feature point are calculated. This is defined by Equation (5). The moments $m_{pq}$ are determined as a weighted sum of the image pixels' intensities within a circle of radius $r$ and centre in the given feature point.

$$m_{pq} = \sum_{x,y} x^p y^q I(x, y) \tag{5}$$

where $x, y$ are the local coordinates within a circle of radius $r$ relative to the detected feature point, $I(x, y)$ is the brightness of the pixel in a given location $(x, y)$.

Then, the orientation of the feature point is determined by Equation (6):

$$\theta = \arctan 2(m_{01}, m_{10}) \tag{6}$$

The second problem is solved by the well-known multi-scale approach. The FAST detector is applied to each level of the image pyramid to improve scale invariance. The image pyramid is built by smoothing the image with an appropriate low-pass filter (usually Gaussian) and then subsampling the image, usually by scaling down by a factor of 2 along each coordinate direction. The resulting image is then subjected to the same procedure, which is repeated multiple times. At each level of the pyramid, exactly the same feature point detection operations as described above are carried out. Non-maximum suppression

can be applied to eliminate multiple detections of the same corners in different scales. The size of the filter mask is dependent on the scaling factor and the number of scales.

### 2.2. Rotated BRIEF Descriptor

The BRIEF descriptor was first presented in the paper [8]. Based on an image patch of $31 \times 31$ pixels around a given point, the algorithm returns a 256-bit binary feature vector (descriptor). It is calculated using a set of binary intensity tests $\tau$—Equation (7).

$$\tau(\mathbf{p}; u_i, v_i) := \begin{cases} 1 & \text{for } I(u_i) < I(v_i) \\ 0 & \text{for } I(u_i) \geq I(v_i) \end{cases} \tag{7}$$

where $\mathbf{p}$ is the smoothed image patch around a given feature point $(u_i, v_i)$, $u_i, v_i$ are the coordinates of sample pairs $\mathbf{p}$ $(u_i \neq v_i)$, $I(w)$ is the brightness of the pixel in a given location $w = (x, y)$.

The descriptor of a feature point is constructed as a $n$-element bit string—Equation (8). The patch $\mathbf{p}$ must be blurred with a Gaussian filter to reduce the influence of noise.

$$f_n(p) := \sum_{i=1}^{n} 2^{i-1} \tau(\mathbf{p}; u_i, v_i) \tag{8}$$

In the original BRIEF descriptor, 256 pairs of points used for the binary intensity test are randomly selected in a corner's neighbourhood according to a Gaussian distribution. However, this descriptor is not resistant to large in-plane rotation. To address this issue, the authors of the paper [8] suggest computing a BRIEF descriptor for a set of rotations for each image patch, but this approach is computationally very expensive. Therefore, in the paper [3] some modifications to the BRIEF descriptor have been proposed in order to make it more efficient and resistant to rotation. The authors have tested two solutions. In the first approach, called steered BRIEF, they rotated a set of $n$ pairs of coordinates of binary tests along a predetermined set of orientations, according to Equation (9). For this purpose, they used angles equal the increments of $2\pi/30$ ($12°$). As a result, they constructed a lookup table of precomputed BRIEF patterns. However, experiments have shown that this approach is not effective as it increases the correlations in the binary tests set.

$$S = \begin{pmatrix} x_1, \ldots x_n \\ y_1, \ldots y_n \end{pmatrix}$$
$$S_\theta = R_\theta S \tag{9}$$

where: $S$ is the set of $n$ binary tests in location $(x_i, y_i)$, $S_\theta$ is the steered version of $S$, rotated set of $n$ binary tests, $R_\theta$ is the rotation matrix corresponding to the orientation of the feature point.

In order to reduce the correlation in the binary tests set, i.e., to get a better quality descriptor, a different solution was proposed. The authors developed a special algorithm using machine learning to select a set of tests that are as uncorrelated as possible to get high diversity and good performance of the descriptor. This algorithm is a greedy search for a set of uncorrelated tests with means near 0.5. As a result, 256 pairs of points are selected, which are then rotated according to the orientation of the feature point (based on Equation (10)) to make it invariant to rotation. This modification is called rotation-aware BRIEF (rBRIEF).

$$x' = x \cdot cos\theta - y \cdot sin\theta$$
$$y' = y \cdot cos\theta + x \cdot sin\theta \tag{10}$$

where: $(x, y)$ is the initial location, $(x', y')$ is the location after rotation, $\theta$ is the orientation of the feature.

Rotation-aware BRIEF, also called rotated BRIEF, is a component of the ORB algorithm. Among its many advantages, worth mentioning are good performance, robustness, and computational efficiency of matching feature points due to its binary structure.

## 3. Related Work

Many feature point detection and description algorithms were proposed in the literature. However, in the unknown space exploration task using Visual SLAM, the ORB algorithm [9–11] is the most widely used. This method for feature point detection has a capability of a high acceleration due to the possibility of parallelising the calculations. This observation is evidenced by a number of publications that describe its hardware implementations using heterogeneous computing platforms, including FPGAs.

Existing hardware architectures can be categorised into non-stream and stream processing. Non-stream processing approaches, such as [9,12], assume the availability of a buffer that stores the image frame. This enables the BRIEF descriptors to be computed only for patches that are centred on the detected FAST corners. On the other hand, stream processing architectures such as [13,14] do not require external memory to store the input video frames and can achieve higher throughput. However, in this approach, where the incoming pixels are processed on-the-fly without being stored in frame buffers, many pipeline and parallel processing elements are needed to keep up with the rate of the incoming pixel stream. The use of algorithms on UAVs' embedded platforms requires very high throughput, speed, and accuracy. Low energy consumption must also be taken into account, which involves a marginal use of memory and additional buffers. For this reason, implementing the ORB algorithm according to the stream processing paradigm is a better option.

The authors of the articles [15–17] focused on the hardware implementation of the ORB algorithm in its basic version, introducing changes only to the mechanism of determining the value of the corner score and adding modules responsible for filtering the obtained feature points. An image pyramid was also used to make the detector robust to scale differences between images. In papers [16,17] systems were proposed that process Full HD images (1920 × 1080 pixels) at 63 fps and 42 fps, respectively. In the articles [9,18], an approach for a more hardware-friendly implementation was proposed. In particular, the BRIEF descriptor was modified. A rotationally symmetric ORB descriptor pattern was used, which drastically reduced the computational complexity and memory usage. In this solution, it was not necessary to rotate the set of binary tests according to the feature point orientation for each image patch, nor to store the BRIEF patterns for all discretised feature point orientations, i.e., 30 sets, each containing 256 pairs of points. Additionally, heap sorting was used to reduce the number of detected feature points, leaving only the most robust ones. The tests were carried out on images with a resolution of 640 × 480 px from the TUM dataset [19]. Also noteworthy is the work of [20], in which the authors presented the concept of a hardware implementation on an FPGA. They performed experiments using functions from the OpenCV library, in which they indicated modifications to be made to speed up the computation.

To compute the BRIEF descriptor of a feature point, two approaches are usually used. The first assumes storing 256 pairs of points and performing the rotation according to Equation (10). However, this requires the calculation of 512 new coordinate values for each feature point. A more popular method is to precompute the rotated BRIEF patterns [3] instead of computing them directly each time. In this approach, the orientation of features is discretised into 30 different values. Then, 30 BRIEF patterns after rotation are precomputed and saved as a lookup table. This reduces the computation cost but increases memory utilisation, while the introduced orientation discretisation deteriorates the accuracy. However, both described approaches are difficult to implement in FPGAs and require very high amount of logical resources, which may also affect energy efficiency. The authors of the work [9] proposed using a 32-fold rotationally symmetric BRIEF pattern (RS-BRIEF), which is more hardware-friendly. They assumed the selection of 2 sets of locations in the neighbourhood around the feature according to the Gaussian distribution.

Each of these 2 sets contained 8 locations, which were rotated by discrete angle values every 11.25 degrees. The final test locations contained 256 pairs of coordinates that were used to determine a binary feature descriptor.

In the works discussed so far, the hardware implementations were adapted to process data with a maximum resolution of Full HD (1920 × 1080 px). Most important parameters of these implementations are gathered in Section 5. In this paper, we describe a hardware implementation of the ORB algorithm for a 4K UHD resolution (3840 × 2160 px) implemented on a ZCU 104 SoC FPGA platform with a Zynq UltraScale+ MPSoC chip from AMD Xilinx. In our ORB implementation, we used the FAST detector and a modified version of the RS-BRIEF descriptor without filtering the points during the subsequent steps. To our best knowledge, this is the first real-time implementation of the ORB algorithm for such a high-resolution video stream.

## 4. The Proposed ORB (FAST+BRIEF) Implementation

Processing a 4K UHD video stream on a SoC FPGA is a significant challenge. The 4K signal contains 4 × more data than Full HD (1920 × 1080 px). Assuming the same frequency (typically 60 fps), the so-called pixel clock needs to be then increased from about 150 MHz to 600 MHz. Using such a clock value in currently available reprogrammable devices is impossible (except for very basic logic elements). Hence, it is necessary to process video data in vector formats—2, 4 or 8 pixels per clock (ppc). This allows to reduce the pixel clock to 300, 150 or 75 MHz respectively [21]. The vector data format implies a different approach during the implementation of individual operations in the ORB (FAST+BRIEF) algorithm. Therefore, the performance of the system described in this paper is not only the result of using a modern SoC FPGA platform, but also the appropriate novel design of particular computational modules. The general scheme of our hardware implementation is shown in Figure 2. This architecture processes the video stream in a fully pipelined manner using AXI4-Stream as the data bus. The input to the module is a stream of greyscale images in 4 ppc format. Note that in the following subsections we present our solution in the 4 ppc format, but the same approach can be applied to 2 ppc or 8 ppc.

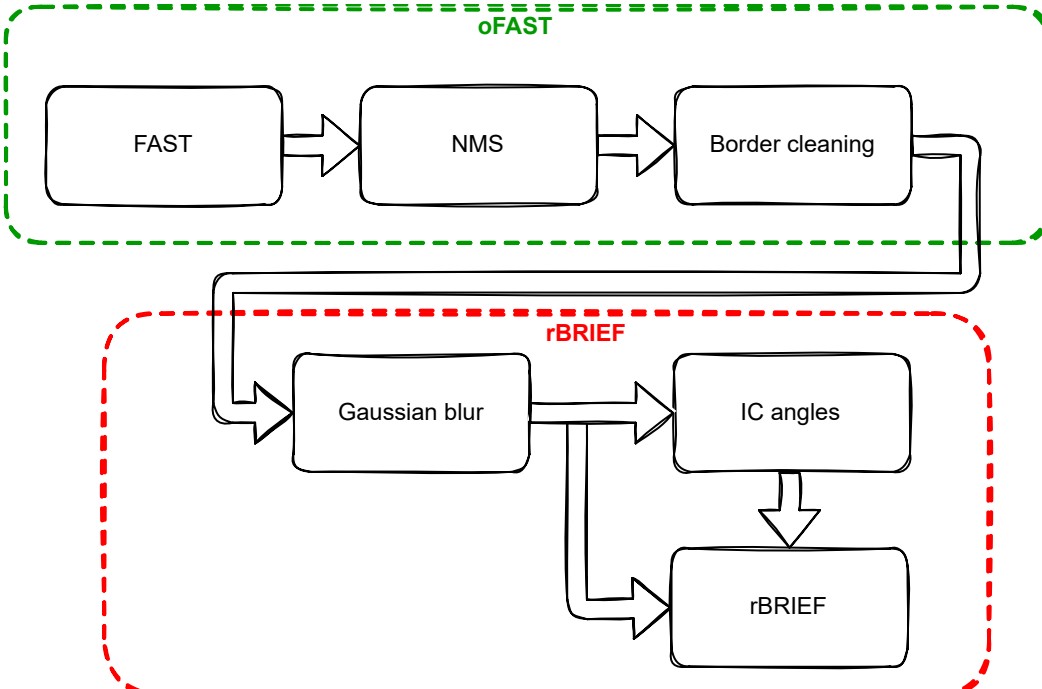

**Figure 2.** Overall scheme of our hardware implementation of the ORB algorithm on a SoC FPGA platform.

As described in Section 2, the ORB algorithm contains an oriented FAST detector and a rotated BRIEF descriptor. Our proposed implementation of the ORB algorithm consists

of several components listed below, which form a modular architecture. Each module can operate independently, like a black box. In our implementation of oriented FAST, we distinguish several stages:

1. Performing the fast accelerated segment test to determine the corners (FAST).
2. Filtration using non-maximum suppression (NMS).
3. Elimination of feature points for which it is not possible to determine the context $31 \times 31$ px (border cleaning).

Similarly, in our implementation of rotated BRIEF, there are several steps:

1. Determination of the context $31 \times 31$ px and blurring with a Gaussian filter.
2. Calculation of the orientation of the feature point (IC angles).
3. Determination of the binary feature descriptor (rBRIEF).

Compared to the originally proposed ORB algorithm, we have decided not to implement the modules that are responsible for determining the Harris score and selecting the best $N$ points. This is motivated by a fully pipelined approach and the need to adapt the algorithm to the hardware implementation. These modifications do not affect the performance of the ORB algorithm in any way, as the computational platform and the processing structure require the processing of all pixels in the image frame. The filtering of points was intended only to reduce the number of calculations in further steps. If necessary, a module can be added that globally sorts the feature points by a corner score. We have also modified the BRIEF descriptor, which we describe in detail in Section 4.3.

In our hardware implementation we use fixed-point operations. Width and precision of data representation in consecutive modules are gathered in Table 1. In each case we use a software model to select numbers' representations as a trade-off between resource utilisation and computing accuracy.

**Table 1.** Fixed-point precision applied at different stages of the ORB algorithm.

| Name | Width | Precision |
|---|---|---|
| Fast Score values | 8 | 0 |
| Gaussian coefficient | 24 | 16 |
| Pixel values after Gaussian filter | 8 | 0 |
| Moments | 21 | 0 |
| Tangent values | 10 | 7 |
| Orientation interval number | 5 | 0 |
| Pattern pairs | 5 | 0 |
| BRIEF descriptor | 256 | 0 |

*4.1. Context Generation in 4K*

The ORB algorithm repeatedly uses operations that exploit pixel values in a close neighbourhood of a given image point. Therefore, appropriately sized contexts created from a 4K video stream are needed. Due to the vector format of the data stream (4 ppc), a typical approach to context generation is not suitable and results in increased resource consumption. In Figure 3 we show a schematic of a context generation with a size of $3 \times 3$ px (for clarity, we present a context with a small size, but it can be generalised to larger ones). The fully pipelined module contains registers to store pixel values ($9 \times 4$ pixels) and delay lines using Block RAM (BRAM).

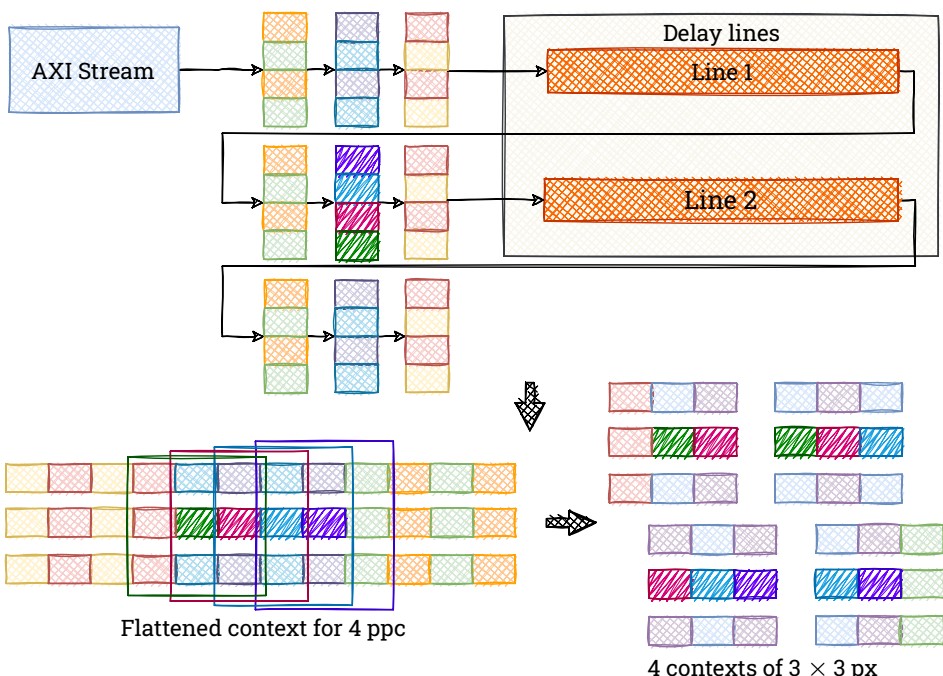

**Figure 3.** Typical scheme for the generation of contexts $3 \times 3$ px for 4 ppc vector format. The dark green, pink, blue and purple pixels are the centres of $3 \times 3$ px contexts.

### 4.2. FAST Feature Detector

Determining feature points using the FAST method requires simultaneous access to pixels from a context of $7 \times 7$ px. To create it, we use the module described in Section 4.1. In Figure 4 we show the scheme of our hardware implementation of the FAST detector.

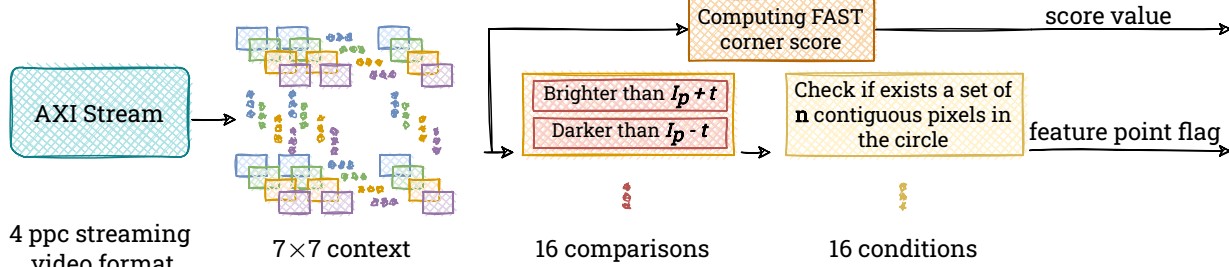

**Figure 4.** Scheme of our hardware implementation of the FAST detector. First, we generate $7 \times 7$ pixel contexts (4 due to the 4 ppc format). Then, for each of them, we test whether its centre is a feature point (feature point flag) and determine its stability (score value). The feature point flag is a signal which indicates that a feature point has been found. It is worth noting the high degree of parallelisation of calculations (e.g., 16 comparisons between centre and circle's points and score value computed simultaneously with the feature point flag), which in addition are performed simultaneously for all generated contexts.

First, each pixel is subjected to a corner check. We perform 16 simultaneous comparisons of the brightness of the candidate point $I_p$ with pixels located on a Bresenham circle of radius 3. The candidate point is considered as a valid feature point if the brightness of $n$ consecutive pixels in the circle is simultaneously lower than $I_p - t$, or simultaneously higher than $I_p + t$, where $t$ is an intensity threshold. We assume $n$ to be 9 and an intensity threshold to be 20, according to the paper [3]. However, these parameters are fully configurable. The resulting vector is compared with all possible combinations representing a 9-bit uninterrupted sequence of ones in a 16-bit vector (also wrapped), e.g., 0011111111100000,

1111000000011111. To get the final result we perform a logical OR operation which combines the obtained results of the comparisons and returns information whether in any of the considered cases at least one correct pattern of bits has appeared. Finding it implies that a given pixel is a possible feature point. Figure 5 shows the way of determining the feature point flag for an exemplary image patch.

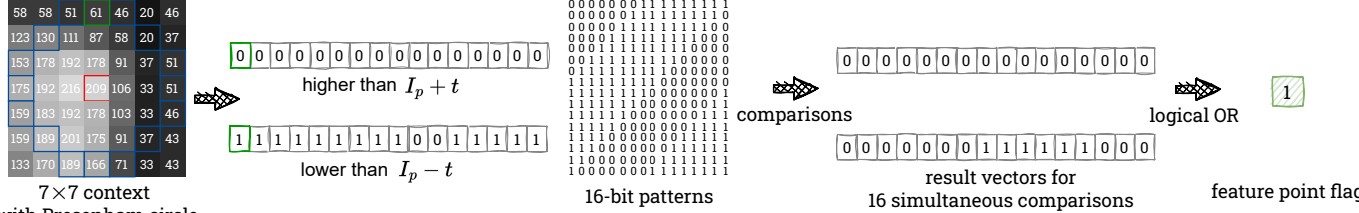

**Figure 5.** Determining the feature point flag for an exemplary image patch. In that case none of the 16 pixels on the Bresenham circle have an intensity above $I_p + t$ (229 for $t = 20$), while 14 of them have an intensity below $I_p - t$ (189). This results in two 16-bit vectors, which are compared with 16 different patterns, each consisting of 16 bits: 9 ones and 7 zeros. The vector representing pixels with intensity higher than $I_p + t$ does not match any pattern, while the second one matches 6 different patterns in total. Therefore, we determine the centre pixel is a feature point (it fulfils the FAST's "lower than" condition).

It is worth noting that in a typical implementation on a CPU the process of finding the feature points is performed in a different way. Firstly, the top and bottom pixels (the only two on the Bresenham circle with $x$ coordinate the same as the centre pixel) are examined—at least one of them must fulfil one of the FAST's conditions. Next, a similar operation is performed for the outer left and right pixels (the only two on the Bresenham circle with $y$ coordinate the same as the centre pixel)—at least one of them must fulfil the same condition as top or bottom pixel. Only after positive verification in both mentioned steps, the rest of the pixels on the Bresenham circle are examined. This method helps to reject many candidate points using relatively few computations (simple comparisons), thus significantly increasing the data processing frequency. However, in the fully pipelined hardware implementation this method cannot work in the same way. We cannot simply reject the candidate, because each pixel must be synchronised with others to ensure an uninterrupted data stream. Therefore, we have decided to compare all pixels at once. Thanks to the FPGA's capabilities (parallel computations), we get the feature point flag just after 3 clock cycles, which is comparable with the CPU's implementation for rejected candidates and significantly lower than in the CPU's implementation for accepted candidates.

The next step is to determine the stability of the selected feature point. Due to the high computational complexity of the Harris corner score, we decide to implement the function used in the OpenCV library—*Fast Score* according to Equation (4). During the computation, we consider all 9-pixel arcs of the Bresenham circle. In each of them, we determine the smallest absolute difference. Then, we choose the largest of the minimum values from all considered arcs. All the described operations are performed in parallel in order to obtain 16 minimum values simultaneously. To determine the minimum and maximum values, we use binary comparison trees. Figure 6 shows the way of computing the Fast Score for an exemplary image patch (the same as in Figure 5). To our best knowledge, this is the first implementation of this module on an FPGA.

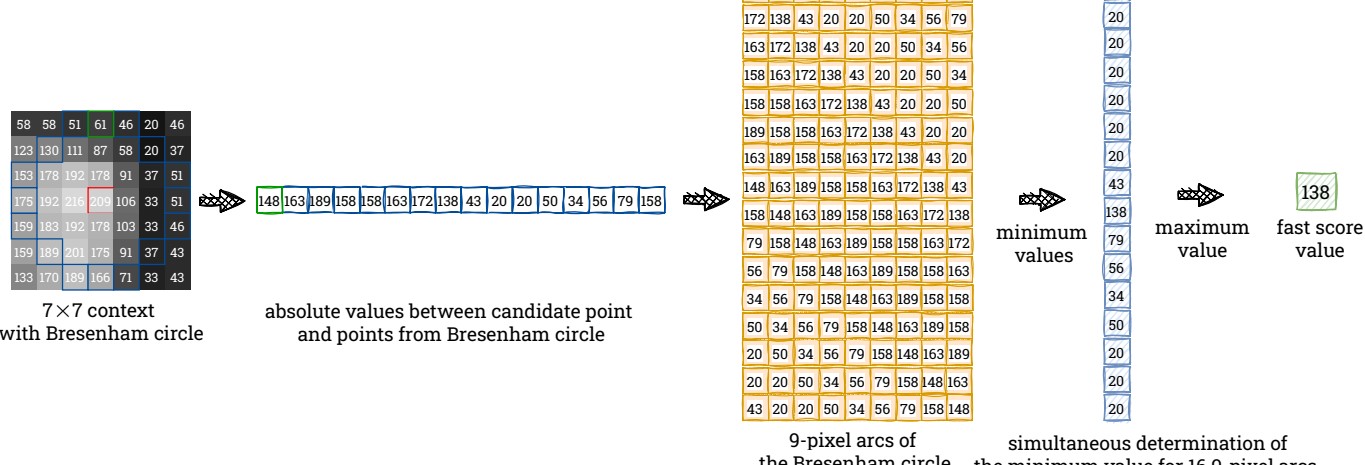

**Figure 6.** Computing *Fast Score* for an exemplary image patch. Firstly, we get 16 absolute differences between the intensity of the centre pixel and the intensities of the pixels on the Bresenham circle. From computed values we form 16 arcs (all possible) with the length of 9 pixels and find the minimum in each arc. *Fast Score* is the maximum from the obtained minimums.

Based on the obtained values, we perform filtering of the determined feature points using non-maximum suppression with context $3 \times 3$ px and thus eliminate points with lower corner scores. The context is created from the corner scores. We pass a non-zero value (input *Fast Score*) to the output if the centre value is the maximum from the entire context and zero otherwise to indicate that a given point is rejected. Then, we also discard feature points that are too close to the edge of the processed image, as we will not be able to create the full context $31 \times 31$ px needed to compute the descriptor. For this purpose, we use coordinate counters and appropriate logical conditions that remove these feature points from the data stream.

The final step is to determine the orientation of the feature point based on the intensity centroid. In the original ORB algorithm, the computing of the image moments needed for this purpose was made before the Gaussian blur and the binary descriptor calculation. In the case of our implementation, we decide to perform the Gaussian blur before the calculation of the image moments and to move these operations to the part directly related to the BRIEF descriptor. Our decision is motivated by the need to reduce the hardware resources used in the FPGA device. We save about 1000 registers, which store a context of $31 \times 31$ px, and 30 Block RAM modules, which store 30 image lines (each containing 2139 pixels). In addition, we eliminate the latency between calculating the orientation of a feature point and determining its description. For both image moments and descriptor calculations, a context of the same size $31 \times 31$ px is used. At the same time, performing Gaussian filtering on the image before determining the orientation does not significantly affect the value of the resulting angle. A similar approach was used in paper [9]. In view of this, it is reasonable to generate the context once on the already blurred image and then use it both to calculate the image moments and to determine the descriptors.

### 4.3. BRIEF Descriptor

As we mentioned in Section 4.2, the first step of our hardware implementation of the BRIEF descriptor is to compute the moments of the image patch around a given point. In Figure 7 we show a schematic of the hardware architecture that we design for this purpose. We first perform a Gaussian blur with a window size of $5 \times 5$ px (as it is implemented in OpenCV), in parallel for 4 pixels in the video stream. Then, we generate contexts of size $31 \times 31$ px. To determine the image moments, we use only the pixels within a Bresenham circle of radius 15 pixels. In Figure 8 we show 4 consecutive contexts $31 \times 31$ px together with the mentioned circles. Note that when processing a 4 ppc data stream in real-time,

it is necessary to perform computations on 4 contexts simultaneously. As can be seen in Figure 8, most of the pixels are common to the four contexts. In view of this, we decide to split the computation into a common part and a residual part belonging to each context separately. Sums in each column and row are calculated in parallel using summation trees. Then we add the resulting sums (common and residual) together to get the total sum of the pixel values in each column and row. Then, they are multiplied by the respective $x$ and $y$ coordinates of the local reference system (with its origin in the point under consideration—the centre of the circle), according to Equation (5). Finally, we sum all the values using summation trees and obtain the moments $m_{10}, m_{01}$ for each context. It is worth noting how we adapt the image moments computation to process the 4 ppc video stream. The proposed division into common and residual parts enables saving a lot of hardware resources that would be consumed by the ordinary multiplication of modules processing a 1 ppc stream.

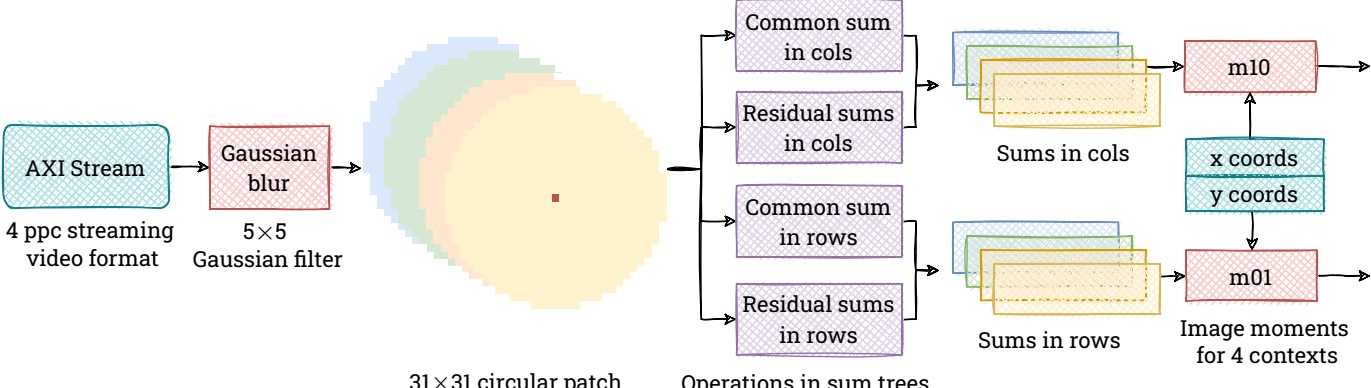

**Figure 7.** Scheme of the hardware architecture used to calculate the image moments of a patch of the image around a given point. First, we blur the image using the Gaussian filter. Next, we analyse successive image patches, which cover circles (radiuses of 15 px) with centres in four neighbouring points. For each circle, we determine the image moments $m_{10}$ and $m_{01}$. For this, we use the sum of the common parts for all circles, thus avoiding performing the same operations repeatedly and saving hardware resources.

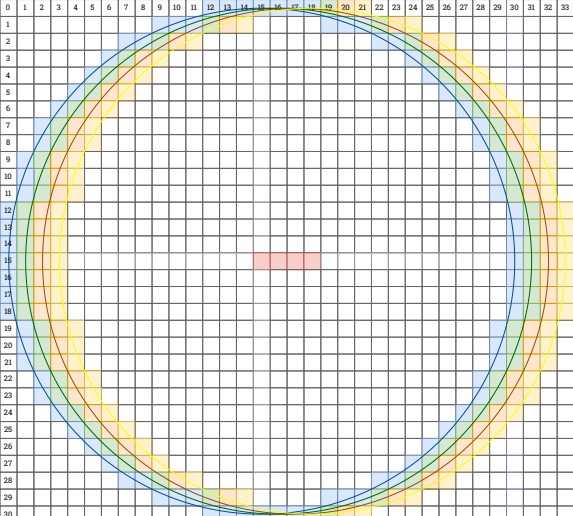

**Figure 8.** Schematic representation of the pixels included in the four consecutive contexts of $31 \times 31$ px. The coloured Bresenham circles mark the circular sections that we use to compute the image moments. In red we indicate the consecutive centres of the circles (the points under consideration).

Then, we need to determine the orientation of the feature point according to Equation (6). In hardware implementation, the most commonly used method for calculating various trigonometric functions is the CORDIC algorithm (Coordinate Rotation Digital Computer). It is based on bit shifting and addition operations, and therefore is considered as a hardware-friendly approach. Its significant drawback is the use of multiple iterations to obtain the best possible precision. For this reason, a solution that involves creating a lookup table to determine the orientation of $m_{01}/m_{10}$ and the signs of $m_{10}$ and $m_{01}$ was also proposed. However, this method uses a divide operation, which is resource-consuming. Therefore, in our implementation, we decide not to compute the exact value of the orientation. Instead of that, we assign the feature point to one of the 32 intervals, for which we transform Equation (6) into an inequality sequence (11):

$$
\begin{aligned}
\tan \theta_i \leq \quad \tan \theta(x,y) &\leq \tan \theta_{i+1} \\
m_{10}(x,y) \tan \theta_i \leq \quad m_{01}(x,y) &\leq m_{10}(x,y) \tan \theta_{i+1}
\end{aligned}
\tag{11}
$$

where: $\theta(x,y)$ is the feature orientation, $m_{10}(x,y), m_{01}(x,y)$ are the image moments, $\tan \theta_i$, $\tan \theta_{i+1}$ are the tangent values of two adjacent intervals to which the orientation will be assigned, 10-bit fixed-point.

Thanks to this, to determine the approximate orientation, we only need simple comparison operations, which are hardware-friendly. In addition, we use the tangent values only for the first quadrant and the sign information of $m_{10}$ and $m_{01}$ in the calculation. It enables further decrease in the utilised hardware resources. In Figure 9 we show a schematic of the mechanism for selecting the appropriate interval for the orientation of the feature.

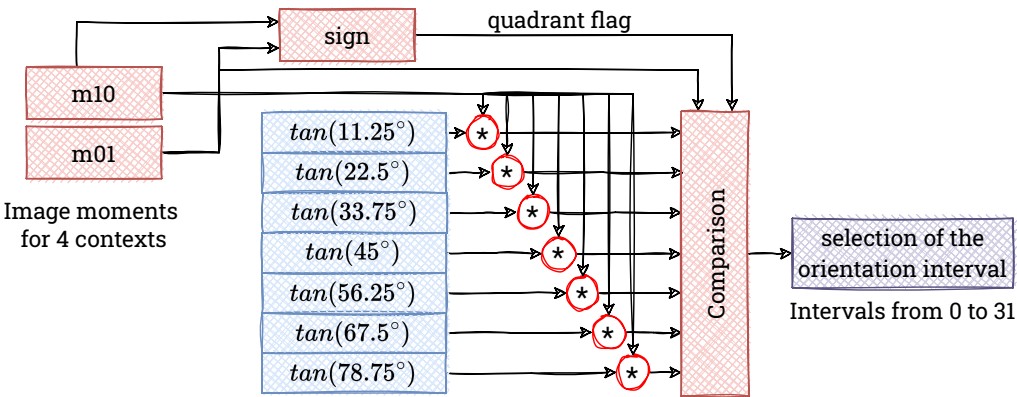

**Figure 9.** Scheme of the mechanism for determining the orientation by assigning a value to an appropriate interval.

In parallel with computing the moments of the image patch and the orientation of the feature point, we determine the BRIEF descriptor using the same context. Following the proposal in the paper [9], we select two sets containing 8 locations and rotate them every 11.25 degrees. In Figure 10 we show a comparison of the pattern for the BRIEF and our implementation of RS-BRIEF. We use the resulting 256-point pairs to determine a 256-bit feature descriptor.

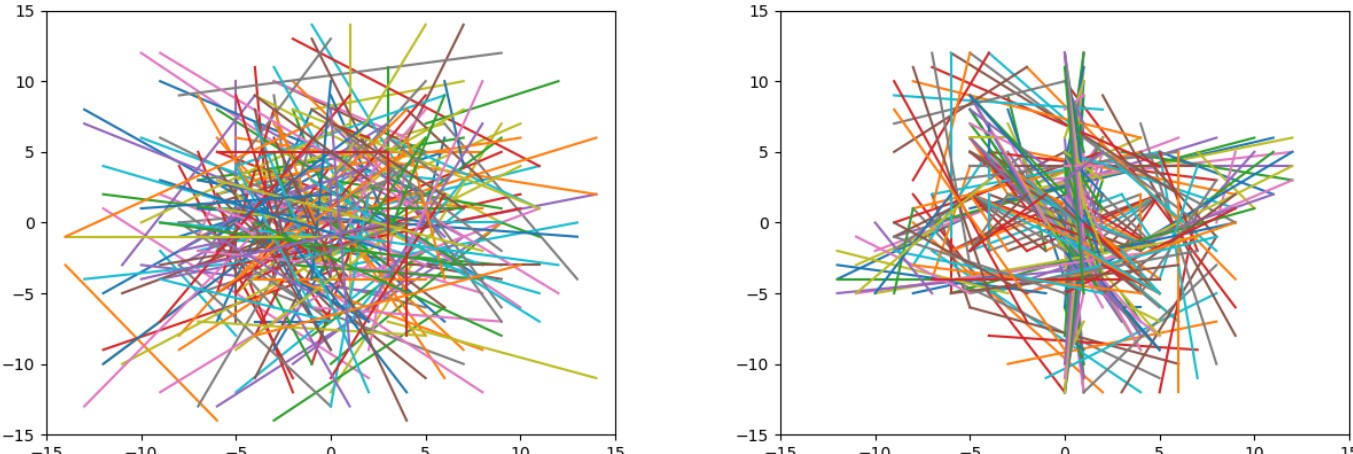

**Figure 10.** Comparison of the test location pattern used in the original BRIEF algorithm (**left**) with that obtained by our RS-BRIEF implementation (**right**).

In the final step, we shift the descriptor according to the orientation of the feature point, which provides the same results as rotating the test locations of RS-BRIEF [9]. Assuming that the orientation of the feature point is $n$, where $n \in \{0, \ldots, 31\}$, the BRIEF vector should be bit-shifted by $8 \times n$ from the beginning of the descriptor to the end. This is exactly how it is realised in our hardware implementation, which we show in Figure 11.

It is worth noting that the use of the RS-BRIEF algorithm may have a negative impact on the quality of the resulting vectors by increasing tests correlation. Nevertheless, trying to implement the BRIEF descriptor in hardware in its original form causes several problems. These primarily concern the computational resources needed to perform the rotation of the test points according to Equation (10). As we mentioned before, this problem is solved in other works by discretising the orientations and storing the precomputed test patterns in a lookup table. However, this leads to a significant increase in the use of memory resources, which is a major drawback of this approach. In view of this, we have decided to use RS-BRIEF. This algorithm reduces the complex implementation of rotation to basic bit-shift operations and leads to a significant reduction in the used hardware resources, while the obtained results are still satisfactory, as we show in Section 5.

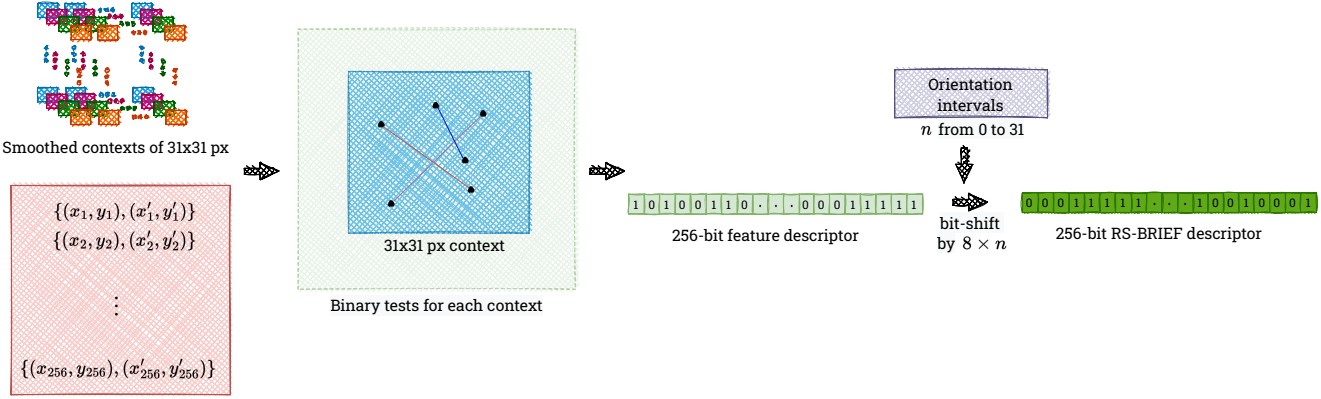

**Figure 11.** Scheme of the RS-BRIEF implementation. We read tests' locations from the memory and perform comparisons between pixels' intensities from $31 \times 31$ px contexts. In this way, we obtain a 256-bit feature descriptor, which we shift according to the orientation.

## 5. Results

We implement the ORB algorithm described in Section 4 on an AMD Xilinx ZCU 104 board equipped with a Zynq UltraScale+ MPSoC EV device, with a quad-core ARM Cortex-A53 application processor, a dual-core Cortex-R5 real-time processor, and a Mali-400 MP2 graphics processing unit. We use SystemVerilog HDL (Hardware Description Language) to describe our implementation in the AMD Xilinx Vivado 2020.2 IDE (Integrated Development Environment). Our system allows real-time processing of 4K resolution (UHD, 3840 × 2160 px) video at 60 frames per second (150 MHz clock). It is capable of detecting and describing feature points in a fully pipelined and parallel manner. However, our algorithm is only rotation-invariant (we use only one scale).

Despite that, we prepare our hardware implementation in a way that allows us to adapt the ORB algorithm to make it scale-invariant in the future. First of all, it is modular, i.e., each functionality depicted in Section 4 is designed as a separate hardware module. If one (or more) of these has to be modified in a multi-scale implementation, this can easily be done by replacing one (or more) modules. Moreover, our implementation is flexible. Each of the aforementioned modules is highly parameterisable, thus allowing straightforward adaptation to different image sizes or vector formats (in general X ppc, where X is a power of 2). The entire system is capable of operating at a maximum clock frequency of 170 MHz (value is estimated with Vivado IDE).

Table 2 shows a summary of the logic resources consumption on the Zynq Ultra-Scale+ ZCU 104 FPGA platform. The use of the resources for the proposed system and the comparison with other FPGA implementations is presented in Table 3. We compare hardware implementations that implement only the FAST detector, only the BRIEF descriptor, the ORB algorithm without the use of an image pyramid, and a fully rotation- and scale-invariant algorithm.

Please note that in our work we use a vector data format (4 ppc) and process 4K UHD @ 60 fps video stream in real-time. In view of this, a direct comparison with existing implementations in terms of used hardware resources cannot be fully meaningful. Despite this, it is worth noting that our work uses a similar number of LUT elements and BRAMs as previous implementations using a multi-level image pyramid. However, we use noticeably more registers (FFs) and DSP blocks. This is due to the hard constraints associated with processing many pixels in a single clock cycle, especially during the determination of the BRIEF descriptor. It should also be emphasised that our implementation uses a relatively small part of hardware resources available in the used FPGA chip. It is therefore possible to extend it with additional components (e.g., multi-level image pyramid) without changing the hardware platform. In particular, our implementation can be used as one of the components of a SLAM system.

**Table 2.** Resource utilisation of our implementation. A significant part of the presented logical resources is consumed by the video pass-through. The high utilisation of DSP blocks is related to the use of the Gaussian filter and the IC module (which performs multiplications for the circular patches from the four contexts of 31 × 31 px.

| Resource | Pass-Through | FAST | BRIEF | ORB | Entire System | Available |
|---|---|---|---|---|---|---|
| LUT | 38,383 | 11,041 | 51,182 | 62,223 (27%) | 100,606 (44%) | 230,400 |
| LUTRAM | 4564 | 386 | 11,951 | 12,337 (12%) | 16,901 (17%) | 101,760 |
| FF | 45,278 | 12,071 | 82,942 | 95,013 (21%) | 140,291 (31%) | 460,800 |
| CARRY8 | 925 | 1056 | 9043 | 10,099 (35%) | 11,024 (38%) | 28,800 |
| BRAM | 7 | 9 | 36 | 45 (14.4%) | 52 (17%) | 312 |
| DSP | 15 | 0 | 668 | 668 (38.7%) | 683 (40%) | 1728 |

**Table 3.** Comparison of our proposed solution with other algorithms reported in scientific papers. A direct comparison of our implementation can only be made with works that also use video pass-through architectures, i.e., [13,14,16,22–24]. In comparison to previous works, we use noticeably more FFs and DSP blocks. This is due to the 4 ppc real-time implementation of the BRIEF descriptor.

| | FPGA | Algorithm | # of LUTs | # of Registers, FFs | BRAM | DSP Blocks | FPS | Freq. [MHz] | Resolution |
|---|---|---|---|---|---|---|---|---|---|
| [25] | AMD Xilinx Zynq-7000 | ORB [+,1] | 4257 | 3187 | 576 Kb | - | 55 | 100 | 640 × 480 |
| [13] | AMD Xilinx ZedBoard | ORB [#] | 9866 | 17,412 | 1.33 Mb | - | 325 | 100 | 640 × 480 |
| [14] | Altera Stratix V | ORB [#,4] | 25,648 | 21,791 | 9.44 Mb | 8 | 67 | 203 | 640 × 480 |
| [26] | Altera Aria V | ORB [+,8] | 206,000 | 231,973 | 8.58 Mb | 449 | 72 | 150 | 1920 × 1080 |
| [27] | AMD Xilinx Kintex-7 | ORB[+] | 80,472 | 112,166 | 35 Kb | 0 | 310 | 100 | 512 × 512 |
| [16] | AMD Xilinx ZedBoard | FAST [#,2,6] | 5700 | 6272 | 1.984 Mb | - | 63 | 148.5 | 1920 × 1080 |
| [22] | Altera Aria V | BRIEF [#,3] | 12,523 | 10,019 | 110 Kb | 0 | 60 | 175 | 1920 × 1080 |
| [12] | AMD Xilinx Ultrascale+ | ORB [+,5] | 28,168 | 9528 | 1.47 Mb | 33 | 108 | 200 | 1920 × 1080 |
| [9] | AMD Xilinx XCZ7045 | ORB [+,6] | 56,954 | 67,809 | 2.73 Mb | 111 | 76 | 100 | 640 × 480 |
| [23] | AMD Xilinx Kintex-7 | ORB [#] | 54,435 | 30,281 | 1.836 Mb | 44 | 161 | 150 | 1280 × 720 |
| [28] | Altera Cyclone V | ORB [+,9] | 5711 | 5453 | 0.3–2.3 Mb | - | 325 | 100 | 1280 × 720 |
| [24] | AMD Xilinx Virtex-7 | ORB [#,6,9] | 71,423 | 49,649 | 3.132 Mb | 285 | 68.8 | 142.8 | 1920 × 1080 |
| [18] | AMD Xilinx ZCU 104 | ORB [+,6,7] | 146,572 | 74,166 | 7.43 Mb | 173 | - | 100 | - |
| **Ours** | **AMD Xilinx ZCU 104** | **ORB [#]** | **100,606** | **140,291** | **6.7 Mb** | **683** | **60** | **150** | **3840 × 2160** |

[#] Stream-based architecture. [+] Non-stream-based architecture. Image is stored in external memory. [1] This work is not rotation-invariant. [2] Only oFAST module is implemented. [3] Only rBRIEF module is implemented. [4] This work uses a 2-level image pyramid. [5] This work uses a 3-level image pyramid. [6] This work uses a 4-level image pyramid. [7] This work uses an 8-level image pyramid. [8] This work uses a 9-level image pyramid. [9] Matching module is also implemented.

In order to verify our hardware implementation, we compare the results with a software model. It is constructed using functions from the OpenCV library and consists of the feature point matching from the FLANN submodule (Fast Library for Approximate Nearest Neighbours), the RANSAC algorithm for finding the inliers, and the homography matrix. As an input to the software model, we use outputs from the OpenCV's ORB as well as outputs from our hardware implementation. In this way, we can compare them in terms of number of inliers, matching rate, rotation error and translation error—results are gathered in Table 4. Please note that the orientation of the feature points varies in the range ±11.25°, according to the modifications used in our algorithm. We use sequences from the Oxford *Affine Covariant Feature* dataset [29] for evaluation. We extract features from each pair of images and attempt to find correspondences through a nearest-neighbour search using the FLANN library (see Figure 12). Then, the number of correct matches is computed by the RANSAC algorithm. We use the ground truth homography to compare with the homography matrix obtained from the Oxford *Affine Covariant Feature* dataset and to compute rotation and translation errors. The rotation error is a measure of the similarity of the rotation matrices (a value of 0 means that they are identical) determined by Equation (12):

$$v = Rodrigues(R_{GT}R^T)$$
$$\epsilon_R = ||v|| \tag{12}$$

where: $R_{GT}$ is the ground truth rotation, $R$ is the estimated rotation, *Rodrigues* is the function `Rodrigues` from OpenCV that converts the rotation matrix to a rotation vector using the Rodrigues transformation [30], $v$ is the rotation vector, $\epsilon_R$ is the rotation error.

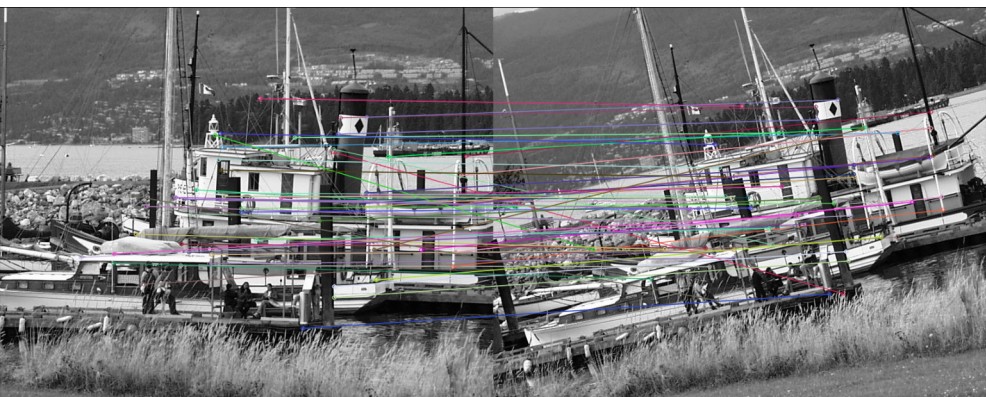

**Figure 12.** Feature matching results using the proposed architecture on images from the *Boat* sequence from the Oxford *Affine Covariant Feature* dataset (the matching part was performed in the CPU).

The translation error measure compares the angle difference between the true rotation and the rotation estimated by Equation (13). The error is expressed in degrees.

$$\epsilon_t = \arccos\left(\frac{t_{GT}t}{||t_{GT}|| \cdot ||t||}\right) \tag{13}$$

where: $t_{GT}$ is the ground truth translation, $t$ is the estimated translation, $\epsilon_R$ is the translation error.

We must emphasise that the results presented in Table 4 indicate that the proposed implementation of the ORB algorithm is correct and the obtained results are similar or better than those of the OpenCV library. Therefore, our module can be successfully used for tasks related to the VSLAM algorithm, for example, for Unmanned Aerial Vehicle platforms. The resulting inconsistencies (e.g., in the number of detected feature points) between our implementation and the functions from OpenCV may be related to minor differences between the two implementations and the use of fixed-point representation.

The number of feature points detected does not have an impact on the error, but a higher number of inliers affects the accuracy of rotation and translation determination.

**Table 4.** Evaluation of the performance quality of our proposed implementation of the ORB algorithm.

| Sequence | # of Image | Implementation | # of Keypoints | # of Matches | # of Inliers | Matching Rate | Rotation Error | Translation Error [°] |
|---|---|---|---|---|---|---|---|---|
| Boat | 1 2 | OpenCV | 507 507 | 45 | 38 | 84% | 0.01679 | 2.06 |
| | 1 2 | Hardware | 872 892 | 118 | 74 | 63% | 0.00116 | 1.50 |
| Bikes | 1 2 | OpenCV | 537 514 | 201 | 199 | 99% | 0.00237 | 6.49 |
| | 1 2 | Hardware | 486 462 | 235 | 221 | 94% | 0.00062 | 4.95 |
| Graffiti | 1 2 | OpenCV | 509 508 | 48 | 41 | 85% | 0.01002 | 1.24 |
| | 1 2 | Hardware | 604 607 | 94 | 65 | 69% | 0.01136 | 1.15 |

Table 5 compares the power consumption of our implementation with other works in the literature that have reported this parameter. The dynamic power consumption of our entire system is 4.3 W, the static one 0.8 W and the total one approximately 5 W. It should be emphasised that our implementation in 4K resolution consumes less power than the implementation using Full HD resolution [26] and is close to the energy value of the work [14].

**Table 5.** Comparison of power consumption. We report total power (unless noted otherwise). It can be seen that due to much higher processed resolution and meeting the real-time requirement, our implementation consumes slightly more power compared to other works. However, this is an acceptable level for potential use on Unmanned Aerial Vehicles.

| | Resolution | FPS | Freq. [MHz] | Energy [mW] |
|---|---|---|---|---|
| [14] | $640 \times 480$ | 67 | 203 | 4559 |
| [26] | $1920 \times 1080$ | 72 | 150 | 5340 |
| [22] | $1920 \times 1080$ | 60 | 175 | 456 |
| [12] | $1920 \times 1080$ | 108 | 200 | 873 |
| [24] | $1920 \times 1080$ | 68.8 | 142.8 | 507 (dynamic) |
| **Ours** | $\mathbf{3840 \times 2160}$ | **60** | **150** | **4278 (dynamic power)** |
| | | | | **764 (static power)** |
| | | | | **5042 (total power)** |

## 6. Conclusions

In this paper, we presented a hardware implementation of the ORB algorithm in a heterogeneous SoC FPGA device. Both of its components—a FAST detector and a BRIEF descriptor —operate in a fully parallel and pipelined manner. We achieved real-time processing of a $3840 \times 2160$ @ 60 fps (150 MHz clock) video stream with an estimated energy consumption of approximately 5 W. The proposed architecture works in the 4 ppc vector format, but it can be adapted to another data format (in general X ppc, where X is a power of 2).

The use of high resolution images allows to capture the fine details of objects and therefore a more accurate description of feature points, which can affect their stability and repeatability. The more precise the detection of feature points, the more accurate the determination of the displacement (rotation, translation). Due to this, for example, we can obtain a very precise trajectory of a UAV moving in an unknown space or generate a better map of the environment. We strongly believe that our work will have a significant impact on improving and developing other methods, such as Visual Odometry (VO) or SLAM in very high resolutions. The proposed architecture is divided into modules that perform individual tasks independently. Its advantage is the possibility to add further

improvements or methods without modifying the entire system. Its modularity also allows the components of the system to be reused for other tasks without any additional readjustments.

As a part of future work, we will implement a scale-invariant version of the algorithm, able to process several scales of the image pyramid in parallel. By designing a modular and flexible architecture, we will achieve this goal by using the ORB module multiple times in parallel. The next step is to perform feature matching and homography estimation to determine the position of the camera, which is another component of the VSLAM system. We also believe that the resource utilisation and the energy consumption can be further optimised. Another interesting research direction will be data fusion from multiple sources—vision camera, event cameras and IMU. This will allow to obtain good performance in very challenging lighting conditions.

**Author Contributions:** The authors made the following contributions to this work: conceptualisation, M.W., H.S. and T.K.; methodology, T.K.; software, M.W. and H.S.; validation, M.W. and H.S.; formal analysis, M.W. and H.S.; investigation, M.W. and H.S.; resources, M.W. and H.S.; data curation, M.W. and H.S.; writing—original draft preparation, M.W. and H.S.; writing—review and editing, T.K.; visualisation, M.W. and H.S.; supervision, T.K.; project administration, T.K.; funding acquisition, T.K. All authors have read and agreed to the published version of the manuscript.

**Funding:** The work presented in this paper was supported by the National Science Centre project no. 2016/23/D/ST6/01389 entitled "The development of computing resources organisation in latest generation of heterogeneous reconfigurable devices enabling real-time processing of UHD/4K video stream".

**Institutional Review Board Statement:** Not applicable.

**Informed Consent Statement:** Not applicable.

**Data Availability Statement:** Not applicable.

**Acknowledgments:** We would like to thank Krzysztof Blachut and Konrad Lis for help with editing the manuscript.

**Conflicts of Interest:** The authors declare no conflict of interest.

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
