# Peer review of "An Efficient Real-Time FPGA-Based ORB Feature Extraction for an UHD Video Stream for Embedded Visual SLAM"

_electronics, doi:10.3390/electronics11142259_

Round 1

Reviewer 1 Report

The paper is well written, but there are a few minor things to be explained or modified.

Please see the comments in the attached.

Author Response

Dear Reviewer,

We are very grateful for the time you have put into this review. We appreciate your comments and did our best to improve the paper. Most of the issues are addressed in the text and the changes are highlighted in blue colour. We also did major improvements to the language used in the paper. Please find below the responses to your particular remarks.

R1.1 [Abstract] How  fast  is  the clock??? Without that, we cannot confirm the real time processing

In the case of our system, we used a 150 MHz clock, which is sufficient for UHD image processing at 60 frames per second assuming the 4 pixel per clock format. We have added this information in the abstract, in Section 1 - Introduction, Section 5 - Results, and Section 6 - Conclusions.

Abstract: Our hardware implementation supports a 4 ppc (pixels per clock)  format (with simple adaptation to 2 ppc, 8 ppc, and more) and real-time processing of a 4K video stream (UHD – Ultra High Definition, 3840 × 2160 pixels) @ 60 frames per second (150 MHz clock)

Introduction: The main contribution of our work includes the implementation of the FAST detector and the BRIEF descriptor for a 4K resolution video stream (UHD – Ultra High Definition, 3840 × 2160 pixels) @ 60 frames per second, 150 MHz clock.

Results: Our system allows real-time processing of 4K resolution (UHD, 3840 × 2160 px) video at 60 frames per second (150 MHz clock).

Conclusions: We achieved real-time processing of a 3840 × 2160 @ 60 fps (150 MHz clock) video stream with an estimated energy consumption of approximately 5 W.

In addition, the maximum system frequency reported by the Vivado tool is 170 MHz for the whole video system (with video input and output) (we added this information in Section 5 - Results).

R1.2 [Oriented FAST and Rotated BRIEF] - add “as”.

We have changed the whole sentence:

Was: 

It performs as well as SIFT on the task of feature detection while being almost two orders of magnitude faster than it, and it is better than SURF, (as) it has better quality of detection and is also more computationally efficient.

Is:

It performs as well as SIFT on the task of feature detection while being almost two orders of magnitude faster and it is better than SURF, as it has higher detection quality and  computational efficiency.

R1.3 [ Oriented FAST Feature Detector] Q: This does not depend on the resolution of the target images ?

As far as we know the method, “n” should not depend on the resolution of the target images. This is due to the design of the FAST detector itself, which compares the brightness of the central pixel with the pixels on a Bresenham circle of a specific (fixed) radius. Hence, the value of n is determined primarily in relation to the circumference of the circle under consideration. Since the radius of the circle is constant (a different value would imply a different detector than FAST, or at least another variant), so also the value of n should remain constant. Additionally, please note that in the original ORB algorithm, the authors assume that feature point detection is performed on a pyramid of images of different scales (resolutions). Despite changing the spatial parameters of the image, both the radius of the circle and the value of n remain constant.

We added this sentence to the article in Section 2.1 - Oriented FAST Feature Detector:

“It may be considered as a general statement (regardless of image resolution), as this depends primarily on the fixed radius of the used Bresenham circle.”

R1.4 [Results] “Can more details of this implementation method be described. This is just saying “modular”.

We have added some more details we mean by saying “modular”.

The proposed ORB (FAST+BRIEF) implementation: Our proposed implementation of the ORB algorithm consists of several components listed below, which form a modular architecture. Each module can operate independently, like a black box.

Results: First of all, it is modular, i.e. each functionality depicted in Section 4 is designed as a separate hardware module. If one (or more) of these has to be modified in a multi-scale implementation, this can easily be done by replacing one (or more) modules. Moreover, our implementation is flexible. Each of the aforementioned modules is highly parameterisable, thus allowing straightforward adaptation to different image sizes or vector formats (in general X ppc, where X is a power of 2).

Conclusions: The proposed architecture is divided into modules that perform individual tasks independently. Its advantage is the possibility to add further improvements or methods without modifying the entire system. Its modularity also allows the components of the system to be reused for other tasks without any additional readjustments. 

R1.5 [Results] “It is better to show which part of the design is the critical path or bottleneck of clock speed?”

To address the comment about defining the critical path of our implementation, we performed an analysis in the Vivado tool. It turned out that the module for determining the orientation interval is capable of operating at a maximum of 170 MHz. This is the slowest module in our hardware implementation (bottleneck). It is responsible for determining the orientation interval number based on the values of the image moments. It is likely that the extended conditional function affects the maximum processing frequency. Still, this frequency is higher than the one our system operates at, which is 150 MHz, and does not affect the processing performance.

In Section 5 - Results we added this sentence to the article:

“The entire system is capable of operating at a maximum clock frequency of 170 MHz (value is estimations obtained with Vivado IDE).”

Tomasz Kryjak

Assistant professor

AGH University of Science and Technology in Kraków

Reviewer 2 Report

English need to be modified by a suitable expert to let the paper convey correct meaning in almost all paragraphs.

The content does not contain any new theoretical contribution. 

There is no architectural contribution as well.

If the authors could provide details of implementation with suitable examples that may add value to the paper and could be useful for a class of readers.

Author Response

Dear Reviewer,

We are very grateful for the time you have put into this review. We appreciate your comments and did our best to improve the paper. Most of the issues are addressed in the text and the changes are highlighted in blue colour. We also did major improvements to the language used in the paper. Please find below the responses to your particular remarks.

R2.1 English need to be modified by a suitable expert to let the paper convey correct meaning in almost all paragraphs.

The article has been thoroughly checked once again, both by us and an external expert. We have made extensive language corrections. We hope that in the current version the text is clearer and easier to read. All changes to the text are marked in blue.

R2.2 The content does not contain any new theoretical contribution. 

If “new theoretical contribution” means the proposal of a new algorithm, then we agree with this remark. However we would like to point out that in the domain of hardware acceleration of algorithms this is not uncommon. For example, authors of the cited papers [9][12][18] also concentrated on hardware implementation of the FAST or ORB algorithms. Here the aim is to adjust it to the used platform and to obtain best performance (speed, low-energy). During the process of analysis and implementation some minor modifications to the algorithm may be also introduced. In our case this was:

- we replaced the time-consuming and resource-intensive calculation of orientation by trigonometric functions with the determination of the interval to which the feature point belongs,

- we adapted the image moments computation to process the 4 ppc video stream,

- we used RS-BRIEF descriptor instead of the original rotated BRIEF descriptor,

- due to the pipelined architecture, we did not perform filtering and selected the best N feature points before determining their description.

We would also like to point out that keeping the hardware implementation similar to the original and well-known versions (like from the OpenCV library) is beneficial, as it allows smoother transfer of algorithms from CPU to FPGA.

R2.3 There is no architectural contribution as well.

We do not fully agree with this remark. We do not introduce a total novel hardware architecture for the ORB module, but based on our previous experience and state-of-the-art articles we propose a version capable of processing in real-time (60 frames per second) an UHD/4K video stream. This required:

- implementation of all the modules with support for the vector format (4 ppc with easy shift to 2 ppc or 8 ppc). Here, please note, that in FPGA implementing 1 ppc (like in all previous papers) is not possible for 4K @ 60 fps video stream - the required clock rate ~600 MHz is just too high,

- performing multiple operations in parallel while maintaining low consumption of logical resources and energy,

- implementation of the multiple contexts generation (for vector format), which is done different then in “classic” 1 ppc approach and in a fully configurable way, i.e. one module for different context’s sizes,

- implementation of the Fast Score computing for the pipelined data stream on the FPGA (additionally in 4 ppc vector format),

- implementation of image patch moments calculation for a context 31 x 31, which extensively uses the vector format and thus reduces the resources utilisation when compared to the naive approach (multiplication of “classic” 1 ppc modules). 

R2.4 If the authors could provide details of implementation with suitable examples that may add value to the paper and could be useful for a class of readers.

We have once more analysed the description of the research and added the following elements:

-  we added this sentence to the article in Section 4.2 - FAST Feature Detector:

“The resulting vector is compared with all possible combinations representing a 9-bit uninterrupted sequence of ones in a 16-bit vector (also wrapped), e.g. 0011111111100000, 1111000000011111. “

-  we added three new schemes using exemplary data, which explain in detail how the calculations are carried out, in Section 4.2 - FAST Feature Detector and Section 4.3 - BRIEF Descriptor:

Figure 5: Determining the feature point flag for an exemplary image patch. In that case none of the 16 pixels on the Bresenham circle have an intensity above Ip+ t (229 for t=20), while 14 of them have an intensity below Ip- t (189). This results in two 16-bit vectors, which are compared with 16 different patterns, each consisting of 16 bits: 9 ones and 7 zeros. The vector representing pixels with intensity higher than Ip+ t does not match any pattern, while the second one matches 6 different patterns in total. Therefore, we determine the centre pixel is a feature point (it fulfils the FAST's "lower than" condition).

Figure 6: Computing Fast Score for an exemplary image patch. Firstly, we get 16 absolute differences between the intensity of the centre pixel and the intensities of the pixels on the Bresenham circle. From the computed values we form 16 arcs (all possible) with the length of 9 pixels and find the minimum in each arc. Fast Score is the maximum from the obtained minimums.

Figure 11: Scheme of the RS-BRIEF implementation. We read tests’ locations from the memory and perform comparisons between pixels' intensities from 31 × 31 px contexts.

In this way, we obtain a 256-bit feature descriptor, which we shift according to the orientation.

Tomasz Kryjak

Assistant professor

AGH University of Science and Technology in Kraków

Round 2

Reviewer 2 Report

The paper could be accepted